# Applications of Virus-Induced Gene Silencing in Cotton

**DOI:** 10.3390/plants13020272

**Published:** 2024-01-17

**Authors:** Yue Tian, Yao Fang, Kaixin Zhang, Zeyang Zhai, Yujie Yang, Meiyu He, Xu Cao

**Affiliations:** 1College of Biotechnology, Jiangsu University of Science and Technology, Zhenjiang 212003, China; shikouqiushan@163.com (Y.T.); fangyao5627@163.com (Y.F.); zhangkaixin202202@163.com (K.Z.); zhai98doc@163.com (Z.Z.); yang1344813698@163.com (Y.Y.); hemy0504@163.com (M.H.); 2Key Laboratory of Silkworm and Mulberry Genetic Improvement, Ministry of Agricultural and Rural Areas, Sericultural Research Institute, Chinese Academy of Agricultural Sciences, Zhenjiang 212018, China

**Keywords:** cotton, virus-induced gene silencing, functional genomics

## Abstract

Virus-induced gene silencing (VIGS) is an RNA-mediated reverse genetics technique that has become an effective tool to investigate gene function in plants. Cotton is one of the most important economic crops globally. In the past decade, VIGS has been successfully applied in cotton functional genomic studies, including those examining abiotic and biotic stress responses and vegetative and reproductive development. This article summarizes the traditional vectors used in the cotton VIGS system, the visible markers used for endogenous gene silencing, the applications of VIGS in cotton functional genomics, and the limitations of VIGS and how they can be addressed in cotton.

## 1. Introduction

Cotton (*Gossypium* spp.) is an important economic crop to the textile industry, one of the world’s largest industries [1]. The genus *Gossypium* contains 52 species. Four of them are cultivated in agricultural production, consisting of two allotetraploids, *G. hirsutum* and *G. barbadense*, and two diploids, *G. arboretum* and *G. herbaceum* [2,3]. They also serve as ideal models to investigate cell differentiation, cell elongation, cell wall biosynthesis, and polyploidization in plants [1].

The lack of accurate genomic information has always been a restriction in cotton breeding. In recent years, the genome sequences of both diploid and tetraploid cottons have been successfully assembled [4]. Technical advances in high-throughput sequencing and bioinformatics analysis have brought a new epoch for the genomic investigation of cotton. Rich genomic resources will not only contribute to the deep understanding of cotton genome evolution and trait domestication but also accelerate the research on functional genomes in cotton [4]. Understanding the function and regulation of cotton genes is an important step in manipulating these genes in agricultural production. Therefore, the development of a fast and cost-efficient tool is urgently required to fill the gap between cotton genomics and functional genomics. At present, agrobacteria-mediated gene transformation is the main method used to acquire transgenic plants. However, tissue cultures and plant regeneration are time-consuming procedures. Moreover, large genome sizes, gene duplication with functional redundancy, polyploidy, and only a few widely used receptors greatly limit the application of gene transformation in cotton [5]. Consequently, virus-induced gene silencing (VIGS) has become a rapid and effective tool in silencing endogenous genes for cotton functional genomics.

VIGS is an RNA interference-mediated high-throughput reverse genetics technique for functional gene analysis in plants [6]. It knocks down gene expression through posttranscriptional gene silencing (PTGS) [7] (Figure 1). VIGS was first used to describe the recovery of viral symptoms in plants after virus infection [8].

Researchers then proved that this is a natural defense mechanism in plants induced by virus infection [9]. Therefore, researchers engineered virus genomes (complementary DNA (cDNA)) into recombinant viral vectors containing sequences of homologous host genes, which can trigger homologous endogenous gene silencing in plants. Thus, VIGS has been widely applied for rapid and large-scale analyses of gene functions and functional genomics in many higher plants, such as *Arabidopsis* [10], tobacco (*Nicotiana benthamiana*) [11], wheat (*Triticum aestivum*) [12], alfalfa (*Medicago truncatula*) [13], tomato (*Solanum lycopersicum*) [14], and poplar (*Populus euphratica*) [15].

In this review, we discuss the VIGS method in cotton, including modifications and applications, especially for genes whose function has been described. We also elaborate on the challenges and restrictions in the use of this method and suggest future directions for its improvement.

## 2. VIGS Vectors for Gene Functional Analysis in Cotton

In the past few decades, various plant viruses have been engineered using the VIGS system for a large number of plant species. As required, many vectors have been developed and applied in the VIGS system. VIGS vectors contain three kinds of vectors: DNA, RNA, and satellite virus vectors. DNA viruses make up only a minority of plant viruses, with large genome structures and limited movement in plants [16]. Single-stranded DNA (ssDNA) viruses belong to the largest known family of plant DNA viruses [16]. Other types of DNA-based viruses, such as African cassava mosaic virus (ACMV) and Cotton leaf crumple virus (CLCrV), have been efficiently converted into VIGS vectors and successfully applied to cassava and cotton plants [17,18]. RNA viruses are the earliest and most widely used viral vector to establish the VIGS system, due to their small molecular weight and high infection efficiency. RNA virus vectors contain Tobacco mosaic virus (TMV) [19], Tobacco rattle virus (TRV) [20], Barley stripe mosaic virus (BSMV) [21], Bean pod mottle virus (BPMV) [22], Potato X virus (PXV) [23], etc. TMV was the earliest vector based on the model RNA virus [19]. A recombinant virus containing the coding sequence of *Nicotiana benthamiana phytoene desaturase* (*NbPDS*) was constructed and used in plants to successfully knockdown *NbPDS* [19]. Among these RNA virus vectors, the TRV-induced gene silencing (TRV-VIGS) system has many advantages, such as a high silencing efficiency, long duration, mild virus symptoms in host plants, and gene silencing in various tissues, resulting in its wide use [7]. Satellite viruses do not induce any plant diseases, and they are usually unrelated to any disease or interference with the true gene-silencing phenotype [24].

To date, two viral vector systems suitable for cotton gene silencing have been reported. One is based on TRV, and the other on CLCrV [25,26] (Figure 2). The TRV vector contains a bilateral positive-sense single-stranded RNA virus, including the RNA1 and RNA2 genomes. RNA1 encodes two replication enzymes and a protein rich in cysteine, which is important for viral replication and movement. Conversely, RNA2 encodes one coat protein and two nonessential structural proteins, which can be deleted to insert alien sequences [7,9]. CLCrV belongs to the genus *Begomovirus*, family *Geminiviridae* [27]. It is known that CLCrV duplicates in the nucleus, producing double-stranded DNA intermediates as templates for additional genome duplication and transcription [28]. CLCrV is a kind of typical bilateral begomovirus consisting of two 2.6 kb circular single-stranded DNA molecules, DNA-A and DNA-B, which share approximately 200 bp of homologous regions known as common regions [18]. The DNA-A component contains four genes that encode the duplication-related protein, transactivator protein, and coat protein. DNA-B contains two proteins, BV1 and BC1. The BV1 protein is a nuclear shuttle protein (NSP), whereas the BC1 protein mediates the intercellular transport of CLCrV DNA via plasmodesmata [29,30]. The CLCrV-based VIGS vector was first reported to silence genes through particle bombardment in cotton [18].

## 3. Visible Markers for Endogenous Gene Silencing in Cotton

To evaluate and monitor the silencing efficiency of VIGS, marker genes, especially those regulating visible phenotype characteristics, are used, including *PDS*, chloroplastos alterados 1 gene (*CLA1*), the green fluorescent protein (*GFP*), and anthocyanidin synthase (*ANS*) [19,31,32,33]. The *PDS* gene is involved in the biosynthesis of carotenoids, and it has been reported in many plant species. It can serve as a visible marker to evaluate the silencing efficiency, as its knockdown in plants can lead to an albino phenotype, due to the lack of chlorophyll [34]. *CLA1* encodes the 1-deoxy-Dxylulose 5-phosphate synthase related to chloroplast development, and it is highly conserved in various plants [35]. The silencing of *CLA1* results in a bleached phenotype [35]. The GFP signal can be easily observed in plant cells, and the silencing efficiency of a modified TRV–GFP vector has been evaluated in various plant species [36]. It has been shown that the infection efficiency of the vector is equal to that of the original TRV vector [36].

In cotton, *CLA1* is widely used as a positive control to determine the silencing efficiency of VIGS [26]. This gene has been silenced in the four cultivated varieties: *G. hirsutum*, *G. barbadense*, *G. arboretum*, and *G. herbaceum* [37,38]. However, not all *G. hirsutum* varieties have an equal response to TRV-mediated silencing, which is an important consideration for future research on gene function [38]. Moreover, compared to *G. hirsutum*, the silencing of *CLA1* is more prominent in *G. arboretum* and *G. herbaceum*, implying that the efficiency of TRV-mediated VIGS may be influenced by the ploidy level and appears to be higher in diploids [38]. Two cotton phenotypic scorable endogenous genes, *PDS* and phytoene synthase (*PSY*), have been tested as positive controls in TRV-based VIGS [39]. The silencing of *PDS* in both *G. hirsutum* and *G. arboretum* showed the photobleaching effect. Meanwhile, knocking down the expression of *PSY* in red leaf cotton resulted in green cotton leaves with no red color patches. These results suggested that *PDS* and *PSY* could be used as positive controls in the VIGS system. However, although the silencing of *CLA1* and *PDS* can lead to visible photobleaching, it can also result in wilting and plant death. Therefore, it cannot be used for gene silencing throughout the entire plant growth period. To address this issue, the cotton pigment gland formation gene (*PGF*), which has been reported to regulate gland formation, has been developed as a new marker gene to track the efficiency of gene silencing in infected tissues [26]. The silencing of *PGF* in cotton does not influence normal growth and development, and the number of glands is highly correlated with the expression level of *PGF* [26,40]; therefore, the *PGF* gene is a perfect marker for the entire cotton growth period. In a previous study, the proanthocyanidin (PA) metabolic-related genes *ANS* and anthocyanidin reductase (*ANR*) were also silenced via VIGS in *G. barbadense*. Transcripts of these two genes were dramatically reduced. The brownish phenotype of the cotton plants emerged from 8 to 10 days post-vaccination in the *ANS*-silenced lines, whereas no visible phenotype difference between the *ANR*-silenced lines and vector control plants was detected [37]. These results indicate that *ANS* and *ANR* serve as mild marker genes for endogenous gene silencing in VIGS.

## 4. VIGS for Studying Abiotic Stress Response in Cotton

Abiotic stresses, such as drought, soil salinity and alkalinity, and extreme temperature stresses, can induce a series of physiological and biochemical reactions and inhibit the normal growth and development of plants [41,42,43]. VIGS has been widely used to study gene function in cotton under abiotic stress (Table 1). The following sections list the application of VIGS to characterize abiotic-stress-related genes in cotton.

### 4.1. Drought Stress Tolerance

Drought is an important limiting factor in cotton production. Over half of the global cotton supply is grown in areas with severe water shortages. Cotton is recognized as drought-resistant. VIGS is a valuable tool for the functional validation of drought-responsive genes in transcriptional profiles under drought stress in cotton.

By integrating genome-wide RNA sequencing (RNA-Seq) and loss-of-function screening using VIGS with a comprehensive biochemical assay, *GhWRKY59* was identified as an important transcription factor that regulates cotton drought stress [44]. Further analysis showed that GhWRKY59 is phosphorylated by a classical mitogen-activated protein kinase (MAPK) cascade composed of GhMAP3K15-GhMKK4-GhMPK6 and that GhWRKY59 positively regulates GhMPK6 activation through the feedback control of *GhMAP3K* expression. Moreover, GhWRKY59 directly regulates the expression of abscisic acid (ABA)-independent *GhDREB2* and drought-responsive genes by binding to their promoters [44]. This study revealed a complete MAP kinase cascade that phosphorylates and activates an important WRKY transcription factor, and it elucidated a regulatory module composed of GhMAP3K15-GhMKK4-GhMPK6-GhWRKY59-GhDREB2, which is involved in regulating the drought response in cotton. Vacuolar HC-ATPase (V-ATPase) is responsible for the deacidification of the cytosol and the excitation of secondary transport in the vacuolar membrane [45]. Various V-ATPase-related genes have been proven to be involved in regulating the plant response to water deficit. However, their roles in the cotton drought response have not been reported. The silencing of *GhVHA-A* in cotton via VIGS decreased the resistance to the drought response and induced a decrease in chlorophyll content and antioxidant enzyme activity, implying that *GhVHA-A* is a candidate gene to enhance resistance to drought stress in cotton [45]. The tubby-like protein (TULP), which is involved in the plant response to various stresses, has rarely been reported in cotton. The expression of *GhTULP30* was activated by drought stress [46]. The knockdown of the expression of cotton *GhTULP30* through VIGS slowed down the rate of stomatal closure under drought stress and reduced the width and length of stomata, which provided a reference and direction for further exploring the role of TULP in cotton [46]. N^6^-methyladenosine (m^6^A) is the most common internal modification in mRNA, and it is believed to be involved in a series of developmental and biological processes. However, the function of m^6^A modification in drought stress is still poorly understood. The silencing of two Ca^2+^ signal-related genes, *GhECA1* and *GhCNGC4*, via VIGS reduced drought tolerance, resulting in a decrease in m^6^A enrichment in silenced lines [47]. This finding revealed a novel posttranscriptional modification mechanism that involves the methylation of m^6^A on the targeted transcripts of the ABA and Ca^2+^ signaling pathway in regulating the cotton drought response.

### 4.2. Salt Stress Tolerance

Soil salinization is one of the most urgent issues worldwide, due to its negative impact on agricultural production. It has adversely affected over 800 million hectares of arable land globally, accounting for over 6% of the world’s total agricultural land [48]. An appropriate salt concentration can serve as a nutritional component to promote cotton growth. However, a high salt concentration results in the early maturity or senescence of leaves, leading to a decrease in cotton yield [43]. The usefulness of VIGS in studying cotton salt stress resistance has also been confirmed.

The plant lipoxygenase (LOX) gene is a member of the non-heme iron-containing dioxygenase family, and it catalyzes the oxidation of polyunsaturated fatty acid to multifunctional oxygenase. The roles of *LOX* genes have been widely investigated in abiotic and biotic stresses; however, their functions in cotton are still poorly understood. The silencing of two cotton LOX genes (*GhLOX12* and *GhLOX13*) through VIGS exhibited significantly higher chlorophyll degradation and a higher accumulation of H_2_O_2_, malondialdehyde (MDA), and proline, thus improving sensitivity to salt stress [49]. The salt overly sensitive (SOS) pathway is conserved in plants, and it is mainly responsible for transporting intracellular Na^+^ out of the cytoplasm to reduce the content of Na^+^ under salt stress conditions [50]. Knocking down the expression of *GhSOS1* via VIGS makes cotton plants more sensitive to salt stress with decreased growth, insufficient root vitality, and increased Na^+^ content and Na^+^/K^+^ ratio in the roots, stems, and leaves [50]. In addition to positive regulatory genes, there are some factors that negatively respond to salt stress in cotton. The drought-induced 19 (Di19) protein is a Cys2/His2 (C2H2)-type zinc finger protein that plays an important role in normal growth and response to abiotic stress in plants. In a previous study, through VIGS, the silencing of *GhDi19-3* and *GhDi19-4* reduced sensitivity to salt stress, with significantly decreased levels of H_2_O_2_, MDA, and peroxidase (POD), while the activity of superoxide dismutase (SOD) dramatically increased sensitivity [51]. Further analysis indicated that the expressions of the Ca^2+^ signal and ABA-related genes were markedly changed, suggesting that *GhDi19-3* and *GhDi19-4* respond to salt stress by participating in Ca^2+^ and ABA signaling. Long noncoding RNAs (lncRNAs) have been proven to be involved in many biological processes and responses to environmental stress. However, the function of the lncRNA response to salt stress in cotton is poorly understood. β-glucosidases (BGLUs) hydrolyze *β*-D-glycosidic bonds and retain the heteromeric configuration that is involved in plant biotic and abiotic stresses [52]. Knocking down the expression of *GhBGLU24-A* via VIGS resulted in a salt-resistant phenotype in cotton while also enhancing plant height and fresh weight [53]. Meanwhile, the lncRNA *TRABA* binds to the promoter of *GhBGLU24-A* to inhibit its expression. Further analysis showed that the GhBGLU24-A interacted with the RING-type E3 ubiquitin ligase GhRUBL, and, by using VIGS, these two genes were both confirmed to be involved in endoplasmic reticulum (ER) stress. Moreover, the function of *GhRUBL* under salt stress was also investigated using the VIGS method, and the silencing of cotton *GhRUBL* increased salt resistance. These findings help to understand the regulatory mechanism of the lncRNA *TRABA* in regulating cotton salt and ER stresses, enabling potential suggestions regarding the development of more resilient crops.

### 4.3. Cold and Heat Stress Tolerance

The extreme temperature fluctuations caused by climate change suppress normal plant growth and threaten crop productivity. Due to global warming, plants are increasingly affected by temperature stress, including cold and heat stresses. Plants have developed sophisticated strategies to quickly sense and effectively respond to temperature stress. The activation of temperature-related genes is important in resisting temperature stress in plants. VIGS is an effective tool to investigate functional genes under temperature stress in cotton.

The members of the P4 subfamily of P-type ATPases are associated with lipid asymmetry between the two lipid lobules of the *Arabidopsis* plasma membrane and are crucial for low-temperature tolerance; however, the functions of P4-ATPases in cotton are still unclear. The silencing of *GbPATP* through VIGS made cotton plants more sensitive to low temperatures and increased the content of MDA with lower catalase (CAT) activity, implying that *GbPATP* acts as an important regulatory factor in cotton low-temperature tolerance [54]. One short-chain alcohol dehydrogenase, *GhSAD1,* which responds to low-temperature tolerance, was identified in a genome-wide association study (GWAS) of 200 cotton materials [55]. Knocking down the expression of a haploid genotype of *GhSAD1* (*GhSAD1^HapB^*) decreased the tolerance of cotton to low temperatures. Further analysis showed that *GhSAD1^HapB^* modulated the activity of the C-repeat binding factor, which regulates the ABA signaling pathway. Moreover, *GhSAD1^HapB^* activated the expression of COLD-REGULATED (COR) genes and increased the number of metabolites related to low-temperature tolerance. These findings improve the understanding of the mechanisms underlying the GhSAD1-mediated low-temperature response in cotton. Natural antisense transcripts (NATs) have been known to be important regulatory factors of gene expression under abiotic stresses in model plants. However, their function under low-temperature stress is poorly understood in crops. *CAN1* was identified from the NATs of the leaves of *G*. *hirsutum* and *G*. *barbadense* under low-temperature stress [56]. VIGS experiments indicated that *CAN1* significantly improved the cold tolerance of cotton, and further investigations showed that *CAN1* regulated the expression of *SnRK2.8* in response to low-temperature stress. These findings indicate the potential of NATs for application in breeding cold-resistant cotton. Mitogen-activated protein kinase kinase kinases (MAP3Ks) regulate various plant biological processes, but their functions in cotton are still unclear. At present, there are few studies examining high-temperature stress in cotton. Knocking down the expression of *GhMAP3K65* through VIGS improved cotton tolerance to heat stress, which provided further insight into the regulatory mechanisms of a Raf-like MAP3K65 protein in cotton [57].

## 5. VIGS for Studying Biotic Stress Response in Cotton

In addition to abiotic stress, several biotic stresses, including pest and pathogen stresses, can also affect the growth and development of plants. Wilt diseases, such as *Fusarium proliferatum* and *Verticillium dahliae*, are recognized as the most damaging environmental stresses in cotton. These diseases can induce serious growth arrest and a loss of yield in cotton. VIGS has been an effective tool to investigate the function of cotton genes under biotic stress (Table 1).

### 5.1. Verticillium Stress Tolerance

*Verticillium* wilt is a serious disease caused by *Verticillium dahliae* wilt. Once infected with *V. dahliae*, the root xylem vessels become blocked, resulting in a decrease in yield and fiber quality [58]. Many functional genes that respond to *Verticillium* stress have been investigated to improve the understanding of the molecular mechanism underlying cotton resistance to *Verticillium*.

Pectin is the main component of the primary cell wall in plants and serves as the main barrier against pathogens. Pectin methylesterases (PMEs) catalyze the demethylation of the pectin galacturonic domain in plant cell walls. Their activity is regulated by PME inhibitors (PMEIs). The silencing of *GhPMEI3* in cotton via VIGS enhanced susceptibility to *V. dahliae* [59]. Glutathione S-transferases (GSTs) have been classified into soluble and microsomal proteins, and they play crucial roles in detoxifying antioxidants and exogenous substances. GSTs protect cells from abiotic or biotic stresses by catalyzing the reduction in tripeptide glutathione (GSH) to bind to various exogenous or endogenous substrates in vivo [60]. The virus-induced silencing of the GST gene (*Gh_A09G1509*) made the resistant cotton cultivar Nongda601 dramatically susceptible to *V. dahliae* [60]. Further analysis revealed that the GST gene was crucial for achieving a subtle balance between the production and scavenging of H_2_O_2_ under *V. dahliae* stress [60]. Suberin acts as a stress-induced resistance barrier in the root cell wall. CYP86A1 encodes the cytochrome P450 fatty acid ω-hydroxylase, which has been reported to be an important enzyme in the biosynthesis of suberin. However, the function of *CYP86A1* in responding to fungi and the mechanisms associated with immune responses are poorly understood. *GbCYP86A1-1* was identified as a disease-resistance-related gene, and the knocking down of *GbCYP86A1-1* through VIGS particularly increased susceptibility to *V. dahliae* in cotton [61]. These results emphasize the function of *GbCYP86A1-1* in defense against fungi and its potential molecular immune mechanisms during this process [61]. Changes in the active structure of the actin cytoskeleton are a common host response to pathogen attack. The role of the cotton actin-binding protein VILLIN2 (GhVLN2) in host defense against soilborne fungus *V. dahliae* wilt was investigated, and the silencing of *GhVLN2* via VIGS reduced the degree of actin filament bundles and interfered with cotton plant growth, leading to the formation of twisted organs and brittle stems, as well as a decrease in cellulose content in cell walls [62]. These results revealed that the regulatory expression and functional transfer of *GhVLN2* promote the regulation of the dynamic remodeling of the actin cytoskeleton in the host immune response to *V. dahliae*. The role of lncRNA in the cotton *V. dahliae* stress response is still unclear. Comprehensive lncRNA profiles of the cotton disease response to defend against *V. dahliae* were constructed, and the virus-induced silencing of two key lncRNAs, *GhlncNAT-ANX2* and *GhlncNAT-RLP7,* in cotton improved tolerance to *V. dahliae* and *Botrytis cinerea,* which may be related to the enhanced expression of *LOX1* and *LOX2* [63]. This is the first description of lncRNA in responding to fungal disease, and it provides new clues for elucidating the mechanism underlying the cotton disease response.

### 5.2. Bemisia Tabaci (Whitefly) Stress Tolerance

Whitefly has caused significant damage to global cotton production. Information on how cotton plants perceive and defend against this destructive pest is limited. RNA-seq was employed to compare two cotton cultivars exhibiting strong resistance and sensitivity to the whitefly. Analysis revealed that the transcriptional response of cotton to whitefly invasion involves transcription factors, genes encoding protein kinases, phytohormone signals, and metabolite synthesis. The virus-induced silencing of *GhMPK3* in cotton led to the inhibition of the MPK–WRKY–jasmonic acid (JA) and ethylene (ET) pathways, and it increased susceptibility to the whitefly [64].

## 6. VIGS for Studying Vegetative Development in Cotton

Cotton vegetative development includes seed germination, root elongation, seedling development, and plant growth. VIGS has been widely used to investigate the functional genes related to vegetative development in cotton (Table 1).

### 6.1. VIGS for Studying Seed Germination in Cotton

Seed germination is a critical stage in the crop life cycle, and it is closely associated with seedling survival rate and crop yield. A low germination rate, emergence rate, and vigor of seeds lead to a decrease in cotton production. Thus, it is of significance to investigate the function and regulation of genes during seed germination in cotton. Recently, an improved VIGS system was developed by simplifying the seed imbibition (Si-VIGS) of the widely used TRV to expand the study of functional genes during germination [65]. Through the functional screening of the cDNA library of germinated cotton seeds, an important regulation gene, *galactose synthase 2* (*GhGOLS2*), for cotton seed germination was discovered. The silencing of *GhGOLS2* in cotton through the Si-VIGS system dramatically suppressed the seed germination process.

### 6.2. VIGS for Studying Root Development in Cotton

Roots absorb water and nutrients supporting plants to respond to environmental stress and significantly affecting many important agronomic traits. However, cotton’s root structure is poorly understood. A GWAS of cotton root traits from 200 upland cotton accessions revealed two candidate genes, *GhTRL1-A05* and *GhPIN8-D04* [66]. The virus-induced silencing of these two genes in cotton resulted in a reduced root length and surface area.

### 6.3. VIGS for Studying Stem Development in Cotton

Stems ultimately originate from the shoot apical meristem (SAM), which is the main organ responsible for transporting molecules from underground roots to the aboveground parts of plants. The virus-induced gene silencing of major DELLA genes particularly increased secondary cell wall (SCW) formation in the cotton stem xylem and phloem [67]. The DELLA protein in cotton has been found to interact with the SCW-related NAC protein, and the silencing of NAC genes via VIGS suppressed the development of SCW by downregulating lignin biosynthesis and deposition [67]. Through the map-based cloning strategy, one key repressor of the formation of the cotton stem trichome *GoSTR* was identified based on an F_2_ segregating population that originated from a cross between TM-1 and J220 [68]. Knocking down the expression of *GoSTR* in J220 and Hai7124 through VIGS led to pubescent stems but no apparent leaf trichome changes.

### 6.4. VIGS for Studying Leaf Development in Cotton

Leaves are derived from the peripheral zone of SAM. The shapes of cotton leaves vary greatly, including okra, Sea-Island, super-okra, and broad leaf shapes, and this is regulated by a multiple allele locus, *L_2_* [69]. Understanding the genetic basis of cotton leaf morphological variation is crucial for improving agricultural production. The *L_2_* loci was identified as a homeodomain leucine zipper (HD-Zip) transcription factor homologous to the *LATE MERISTEM IDENTITY1* (*LMI1*) gene (*GhOKRA*) [69,70]. The silencing of *GhOKRA* through VIGS resulted in a change in phenotype from okra to broad leaf. This study provides a theoretical direction for breeders to generate a superior cotton ideotype. Ethylene accumulates with the senescence process, and it is the main accelerant of leaf senescence. ETHYLENE INSENSITIVE3 (EIN3), as the core transcriptional enhancer, activates the expression of extensive downstream genes during leaf senescence. Cotton *LINT YIELD INCREASING* (*GhLYI*) belongs to the *EIN3-LIKE 1* (*EIL1*) gene, which acts as an ethylene signal response factor and a positive regulator of senescence [71]. *GhLYI* directly binds to the promoter of *SENESCENCE-ASSOCIATED GENE 20* (*SAG20*) to activate its expression. The silencing of *GhSAG20* via VIGS in cotton can delay leaf senescence [71].

## 7. VIGS for Studying Flowering in Cotton

The role of genes that regulate the development of anther and fertility is still yet to be determined. The silencing of acyl-CoA N-acyltransferase (*GhACNAT*) through VIGS in cotton led to indehiscent anthers, reduced filaments and stamens, and plant sterility [72]. Further analysis revealed that *GhACNAT* played a critical role in controlling fertility by regulating lipid biosynthesis and the JA biogenesis pathways. Glycerol-3-phosphate acyltransferases (GPATs) are crucial for various biological processes, such as male fertility, and they have been widely investigated. However, their exact role and potential regulatory mechanisms in cotton anther development are poorly understood. *GhGPAT12/25* has been confirmed to regulate tapetum degradation, anther cuticle formation, and pollen wall development [73]. Further investigation showed that the expression of *GhGPAT12/25* might be activated by *GhMYB80s* in the regulation of male fertility. The virus-induced gene silencing of *GhMYB80s* in cotton exhibited a dramatic reduction in male fertility [73]. These findings provide insights into the regulatory mechanism of cotton anther development. The use of male sterility plays an important role in improving cotton yield and fiber quality. A complete male sterile line (*ms_5_ms_6_*) is widely used in the development of hybrid cotton globally. Duplicate mutations of *GhCYP450* genes, which encode the cytochrome P450 protein, are responsible for causing male sterility in the cotton *ms_5_ms_6_* accession [74]. The suppression of *GhCYP450* in cotton via VIGS resulted in shorter filaments and no mature pollen grains.

## 8. VIGS for Studying Fiber Development in Cotton

The process of cotton fiber development takes a long time; VIGS could also function in this process. The silencing of the *KATANIN* gene through VIGS exhibited shorter cotton fibers and an increased weight ratio of seed oil to endosperm [75]. On the contrary, the suppression of *WRINKLED1* expression led to an increase in fiber length but a decrease in the content of oilseed, indicating the potential of increasing fiber length by redistributing carbon flow. These findings provide evidence that the TRV-VIGS system can be used for rapid functional analysis of genes associated with cotton fiber development. A membrane lipid analysis was employed in wild-type cotton and fuzzless–lintless mutant fiber cells and ovules, and the content of phosphatidylinositol (PI) was found to be significantly higher in fiber cells. The genes encoding fatty acid desaturases (Δ*^15^GhFAD*), PI synthase (*PIS*), and PI kinase (*PIK*) were expressed in a fiber-preferential manner [76]. The silencing of *GhD15FAD*, *GhPIS*, or *GhPIK* via VIGS resulted in shorter cotton fibers. This study provides a foundation for a deeper analysis of the roles of PI and PIP in regulating cotton fiber development. The *MYBMIXTA-like* (*MML*) transcription factors *GhMML3_A12* and *GhMML4_D12* were identified using the map-based cloning strategy; it is interesting that *GhMML3_A12* is arranged in tandem with *GhMML4_D12* [77,78]. The virus-induced gene silencing of these two genes can inhibit cotton fiber initiation, implying that they play crucial roles in fiber development in cotton. Phytohormones play vital roles in plant growth and development. However, the function and regulatory mechanisms of phytohormones in controlling cotton fiber secondary cell wall formation are largely unknown. The silencing of the auxin response factors *GhARF7-1* and *GhARF7-2* through VIGS in cotton inhibited fiber elongation, and the length of mature fibers in silenced cotton plants was shorter than in controls; additionally, mature fiber cell wall thickness was apparently less than that of controls [79].

**Table 1 plants-13-00272-t001:** Descriptions of functional genes proved through VIGS in cotton.

Gene	Category	Function	Reference
*GhMAP3K15*	Response to drought stress	Positive regulator of drought stress	[44]
*GhMKK4*		Positive regulator of drought stress	[44]
*GhMPK6*		Positive regulator of drought stress	[44]
*GhWRKY59*		Positive regulator of drought stress	[44]
*GhDREB2*		Positive regulator of drought stress	[44]
*GhVHA-A*		Positive regulator of drought stress	[45]
*GhTULP30*		Positive regulator of drought stress	[46]
*GhECA1*		Positive regulator of drought stress	[47]
*GhCNGC4*		Positive regulator of drought stress	[47]
*GhLOX12*	Response to salt stress	Positive regulator of salt stress	[49]
*GhLOX13*		Positive regulator of salt stress	[49]
*GhSOS1*		Positive regulator of salt stress	[50]
*GhDi19-3*		Negative regulator of salt stress	[51]
*GhDi19-4*		Negative regulator of salt stress	[51]
*GhBGLU24-A*		Negative regulator of salt stress	[53]
*GhRUBL*		Negative regulator of salt stress	[53]
*GbPATP*	Response to temperature stress	Positive regulator of low-temperature stress	[54]
*GhSAD1*		Positive regulator of low-temperature stress	[55]
*CAN1*		Negative regulator of low-temperature stress	[56]
*SnRK2.8*		Positive regulator of low-temperature stress	[56]
*GhMAP3K65*		Negative regulator of high temperature stress	[57]
*GhPMEI3*	Response to verticillium stress	Positive regulator of verticillium stress	[59]
*GhGST*		Positive regulator of verticillium stress	[60]
*GbCYP86A1-1*		Positive regulator of verticillium stress	[61]
*GhVLN2*		Negative regulator of verticillium stress	[62]
*GhlncNAT-ANX2*		Positive regulator of verticillium stress	[63]
*GhlncNAT-RLP7*		Positive regulator of verticillium stress	[63]
*GhMPK3*	Response to whitefly stress	Positive regulator of whitefly stress	[64]
*GhGOLS2*	Seed germination	Positive regulator of seed germination	[65]
*GhTRL1-A05*	Root development	Positive regulator of root length and surface area	[66]
*GhPIN8-D04*		Positive regulator of root length and surface area	[66]
*DELLA*	Stem development	Negative regulator of stem secondary cell wall formation	[67]
*NAC*		Positive regulator of stem secondary cell wall formation	[67]
*GoSTR*		Negative regulator of stem trichome formation	[68]
*GhOKRA*	Leave development	Regulator of leaf shape	[69,70]
*GhLYI*		Positive regulator of leaf senescence	[71]
*GhSAG20*		Positive regulator of leaf senescence	[71]
*GhACNAT*	Flowering	Positive regulator of fertility	[72]
*GhMYB80s*		Positive regulator of male fertility	[73]
*GhCYP450*		Positive regulator of male fertility	[74]
*KATANIN*	Fiber development	Positive regulator of fiber length	[75]
*WRINKLED1*		Negative regulator of fiber length	[75]
*GhD15FAD*		Positive regulator of fiber length	[76]
*GhPIS*		Positive regulator of fiber length	[76]
*GhPIK*		Positive regulator of fiber length	[76]
*GhMML3_A12*		Positive regulator of fiber initiation	[77]
*GhMML4_D12*		Positive regulator of fiber initiation	[78]
*GhARF7-1*		Positive regulator of fiber secondary cell wall formation	[79]
*GhARF7-2*		Positive regulator of fiber secondary cell wall formation	[79]

## 9. Limitations of VIGS and How They Can Be Addressed in Cotton

VIGS is one of the most promising and effective tools to investigate functional genes in cotton. Its main advantage is the ability to generate rapid phenotypes without the need for stable plant transformation. VIGS is inexpensive compared with other tools, such as T-DNA, transposon insertion technology, and genome editing methods like the CRISPR–Cas system. However, the VIGS method still has several limitations.

Firstly, the gene silencing duration of VIGS is generally species-specific. Although the VIGS method system has a high silencing efficiency during the cotton seedling stage, its efficiency dramatically decreases after flowering. Therefore, in order to make this method suitable for gene function research during the flowering or fiber elongation stage, a friction inoculation method was employed to prolong the period of target gene silencing [26]. Moreover, the improved system can prolong the period of VIGS in other plants [80,81]. The *CLA1* gene was used as a positive control in the improved system in cotton. First, the *Agrobacterium* strain carrying the *CLA1* silencing vector was injected into *Nicotiana tabacum*, and then newly grown tobacco leaves exhibited a macular phenotype two weeks later [26]. Next, about 1–3 g of the macular phenotype tobacco leaves was collected and placed in a mortar, followed by a small amount of phosphate-buffered solution (PBS), and ground into juice. The cotton leaves were inoculated and stained with 500-mesh emery paper. After dipping a grinding rod in the juice and gently rubbing it on the cotton leaves, the leaves displayed slight wounds, but epidermal cells remained undamaged. The newly grown cotton leaves exhibited whitening after two weeks in a greenhouse [26]. A real-time fluorescence quantitative PCR analysis revealed that *CLA1* expression was dramatically reduced. These findings indicate that the improved friction inoculation method in cotton can extend the silencing period of target genes.

Secondly, the differential penetrance of the phenotype in vegetative and reproductive tissues is another challenge of the VIGS method, which requires the phenotype screening of a large number of plants. To date, research has revealed that the silencing effect is usually regional, dividing the entire plant or being limited to the gene-silencing regions of plants with few successive nodes [80,82]. Therefore, an appropriate selection of positive controls is essential for applying the VIGS system in various tissues during the cotton development stage. As an internal reference gene, *PGF* could reflect the silencing efficiency of the target gene in real time based on changes in the number of glands in the newly grown tissue of cotton plants, including leaves, stems, buds, sepals, and bolls [26]. Moreover, the *PGF* gene is one of the best visible markers for tracing the entire growth period of cotton. Therefore, *PGF* is the perfect candidate gene for evaluating the silencing efficiency in both vegetative and reproductive tissues throughout cotton growth.

Furthermore, changes in environmental conditions, including lighting, temperature, and humidity, can influence the silencing efficiency in the process of abiotic stress treatment [83,84]. The silencing efficiency is decreased at a high temperature due to reduced virus reproduction [85]. This can be resolved by pre-validating virus proliferation and keeping plants inoculated with VIGS vectors under optimal environmental conditions until silencing after abiotic stress treatment.

## 10. Conclusions and Future Prospects

As is well known, the transgenic expression of target genes is the most popular and effective way to investigate functional genes. However, due to the genotypic dependence of cotton genetic transformation, it is a time-consuming and laborious task that is sometimes difficult to employ. VIGS is a feasible method for quickly verifying gene function, including abiotic and biotic stress responses, and vegetative and reproductive development in cotton (Figure 3).

Improvement in the inoculation method could expand the application of VIGS in cotton. A leaf injection is traditionally used. However, this restricts the application to the early stage of cotton development, which is important for seedling survival and crop yield. Recently, through the simplified seed imbibition of TRV, an improved VIGS system was developed to expand functional gene investigations during the germination of cotton seeds [65]. This system has been validated by suppressing the seed-germination-related gene *GhGOLS2* in cotton. Moreover, the efficient duration of virus-induced silencing and applicability in several cotton varieties has been successfully estimated. This finding allowed for the establishment of a novel VIGS system to study functional genes during the cotton seed germination and early seedling stages, expanding the application scope of VIGS and promoting research on cotton functional genomics. In addition, the findings and applicability of the *PGF* gene demonstrate its potential as a positive control for evaluating and tracking the efficiency of VIGS in cotton [26,40]. The silencing of *PGF* in cotton has no effect on normal growth and development. Moreover, a change in the gland number is highly associated with the expression level of the *PGF* gene. Therefore, it is a perfect indicator to trace the silencing efficiency through the entire growth period in cotton. Additionally, the method of friction inoculation has been developed and employed to prolong the silencing efficiency of VIGS in cotton [26], aiding in the study of functional genes during both the vegetative stage and reproductive stage. The improved VIGS method has become a powerful tool for the rapid analysis and investigation of unknown gene functions in cotton.

Collectively, these studies serve as crucial references in the improvement and application of VIGS in cotton. However, due to the transient effect and non-inheritable nature, the VIGS method cannot be used for breeding purposes. Recently, it has been proposed that high-pressure double-stranded RNA (dsRNA) spraying directed at the nucleus can produce stable RNA-directed DNA methylation (RdDM) [86,87]. These new findings provide enormous potential for applications of VIGS-based technology in cotton.

## Figures and Tables

**Figure 1 plants-13-00272-f001:**
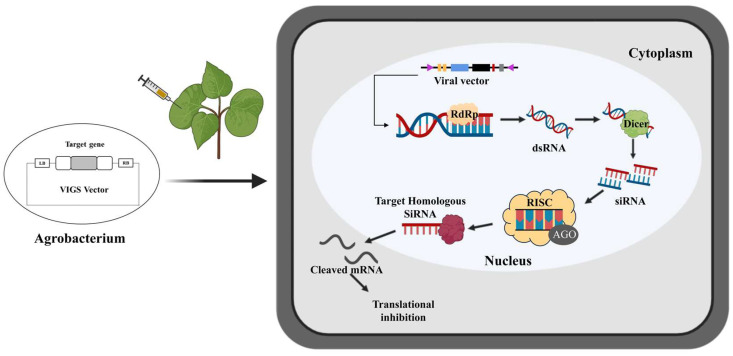
Molecular mechanism of virus-induced gene silencing viral vectors injected into cotton through an agrobacterium carrying the targeted gene. The target gene is fused into the VIGS vector and transformed into *Agrobacterium*. The strains enter the cotton leaf through injection. After infection, T-DNA containing the viral genome is transcribed by the cotton RNA polymerase. With the help of RNA-dependent RNA polymerase (RdRP) (yellow), the single-stranded RNA (ssRNA) viral transcripts produce double-stranded RNA (dsRNA). The dsRNAs are further cleaved into small (21–24 nt) short interfering RNAs (siRNAs) by a dicer (green). Then, amplified siRNAs, together with the AGO protein, form an RNA-induced silencing complex (RISC). RISC uses these siRNAs to accurately establish homologous RNAs in cells, which triggers the endo-nucleolytic cleavage and translational inhibition of the cognate target mRNA, thereby producing PTGS. The single-stranded siRNAs are amplified and propagated as a mobile silencing signal throughout the plant, leading to target gene silencing in plant organs far from the infected site.

**Figure 2 plants-13-00272-f002:**
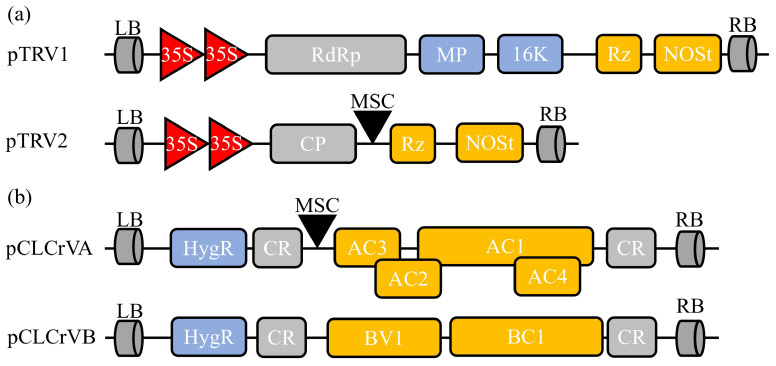
Schematic diagram of two common VIGS vectors used in cotton. (**a**) Tobacco rattle virus (TRV)-derived silencing vector contains pTRV1 and pTRV2. LB, left border; RdRp, RNA-dependent RNA polymerase; MP, movement protein; 16 K, 16 Kd protein; Rz, self-cleaving ribozyme; NOSt, NOS terminator; RB, right border; CP, coat protein; MCS, multiple cloning site. (**b**) Cotton leaf crumple virus (CLCrV)-based silencing vector includes pCLCrVA and pCLCrVB. pCLCrVA consists of four genes (AC1–AC4), and pCLCrVB consists of two genes (BV1 and BC1). The genes are flanked by common regions (CRs), including the origin of replication.

**Figure 3 plants-13-00272-f003:**
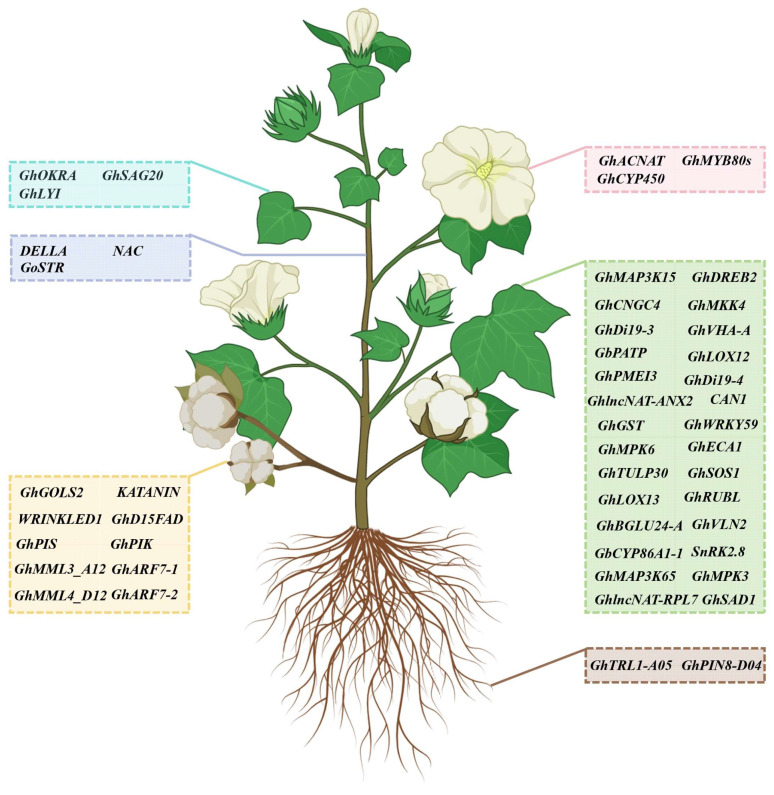
A representation of functional genes validated via VIGS in cotton. The blue box represents genes that function in the regulation of leaf development; the purple box indicates genes that function in the regulation of stem development; the pink box represents genes that function in the regulation of flower development; the yellow box indicates genes that function in the regulation of fiber and seed development; the brown box represents genes that function in the regulation of root development; the green box indicates genes that respond to abiotic and biotic stresses.

## Data Availability

Not applicable.

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
