# Peer review of "Applications of Virus-Induced Gene Silencing in Cotton"

_plants, 2024, doi:10.3390/plants13020272_

Round 1
Reviewer 1 Report
Comments and Suggestions for Authors
The article, "Applications of virus-induced gene silencing (VIGS) in cotton" is well summarized. The article summarized the importance of the VIGS technique in plants, specifically Cotton. Cotton is an important crop, and many countries' economies are based on this crop, but the production is also greatly affected in recent times. This needs time to review and concise all the studies related to VIGS in one document to decide the direction of further studies.
The article explains the VIGS system, its application, and its limitations for better understanding. I have observed some references are very old 1995, 1998 etc. I would recommend the authors use the latest references not later than 2015, although the old references are important, try to use the most recent one.
Comments on the Quality of English LanguageEnglish is fine
Author Response
Thanks so much for the suggestions that we should use the latest references not later than 2015, since part of the references are very old. We have revised part of the references as far as possible. Now, reference 9 to 14, reference 18, 20, 21, 27, 28 have been updated. Even though, like reference 8 and 19, has not been changed due to they were the first report about VIGS and application in plants. In this review, we have summarized the VIGS system, its application, and its limitations for better understanding, we believe the history of the technology itself is also important, so we just retain reference 8 and 19.
Reviewer 2 Report
Comments and Suggestions for Authors
See attached file

See attached file
Author Response
- Line 11, we have changed ‘developed into’ to ‘become’.
- Line 12, we have added ‘has been successfully’.
- We have deleted Line 14 to 17 ‘Moreover, the recent study associated with VIGS system in cotton also expands the application range of the method itself. Therefore, review of technical progresses related to the VIGS method in cotton needs to be timely updated’.
- We have deleted Line 20 to 21 ‘which provides a better understanding and recommendations for investigations of gene function in cotton’.
- From Line 25 to 28, we have changed to ‘Cotton (Gossypium spp.) is an important economic crop to the textile industry, one of the world’s largest industries [1]’. In addition, we have used a new reference that associated with the meaning.
- From Line 28 to 29, we have changed to ‘The genus Gossypium contains 52 species. Four of them are cultivated during agricultural production’.
- Line 31, we have changed the reference 3 to ‘Flagel, L.E., Wendel, J.F., Udall, J.A. Duplicate gene evolution, homoeologous recombination, and transcriptome characterization in allopolyploid cotton. BMC Genomics 2012, 13, 302.’
- We have deleted Line 31 to 33 ‘Among them, cotton fibers produced by hirsutum and G. barbadense account for about 98% of the word’s textile fibers due to these two species are the most widely cultivated worldwide’.
- Line 35, we have added the reference 1.
- From Line 36 to 37, we have changed to ‘The lack of accurate genomic information has always been a restriction in cotton breeding’.
- From Line 48 to 49, we have changed to ‘However, tissue cultures and plant regeneration are time-consuming procedures.’
- From Line 54 to 59, we have changed to ‘VIGS is an RNA interference-mediated high-throughput reverse genetics technique for functional gene analysis in plants [6]. It knocks down gene expression through posttranscriptional gene silencing (PTGS) [7] (Figure 1). VIGS was firstly used to describe the recovery of viral symptoms in plants after virus infection’
- In figure1, we have revised the figure 1 and rewrite the legend to ‘Molecular mechanism of virus-induced gene silencing viral vectors injected into cotton through an agrobacterium carrying the targeted gene. Target gene is fused into VIGS vector and transform into Agrobacterium. The strains enter cotton leaf through injection. After infection, T-DNA containing the viral genome is transcribed by the cotton RNA polymerase. With the help of RNA-dependent RNA polymerase (RdRP) (yellow), e single-stranded RNA (ssRNA) viral transcripts produce double-stranded RNA (dsRNA). The dsRNAs are further cleaved into small (21–24 nt) short interfering RNAs (siRNAs) by a dicer (green). Then, amplified siRNAs, together with the AGO protein, form an RNA-induced silencing complex (RISC). RISC uses these siRNAs to accurately establish homologous RNAs in cells, which triggers the endo-nucleolytic cleavage and translational inhibition of the cognate target mRNA, thereby producing PTGS. The single-stranded siRNAs are amplified and propagated as a mobile silencing signal throughout the plant, leading to target gene silencing in plant organs far from the infected site.’
- From Line 75 to 81, we have changed to ‘Researchers then proved that it is a natural defense mechanism in plants induced by virus infection [9]. Therefore, researchers engineered virus genomes (complementary DNA (cDNA)) into recombinant viral vectors containing sequences homologous host genes, which could trigger homologous endogenous gene silencing in plants.’
- From Line 86 to 89, we have changed to ‘In this review, we discuss the VIGS method in cotton, including modifications and applications, especially for genes whose function has been described. We also elaborate on the challenges and restrictions in the use of this method and suggest future directions for its improvement’.
- From Line 104 to 107, we have changed to ‘A recombinant virus containing the coding sequence of Nicotiana benthamiana phytoene desaturase (NbPDS) was constructed and used to plants to successfully knockdown NbPDS.’
- From Line 109 to 110, we have changed to ‘resulting in its wide use.’
- Line 114, we have changed to ‘and the other on CLCrV.’
- Line 127, we have changed to ‘The BV1 protein is a nuclear shuttle protein (NSP).’
- In figure 2. We have replaced reference 20 to ‘Singh, A.K., Ghosh, D., Chakraborty, S. Optimization of Tobacco rattle virus (TRV)-based virus-induced gene silencing (VIGS) in tomato. Methods Mol. Biol. 2022, 2408, 133-145.’
- From Line 141 to 143, we have changed to ‘those regulating visible phenotype characteristics used, including PDS, chloroplastos alterados 1 gene (CLA1), the green fluorescent protein (GFP), and anthocyanidin synthase (ANS).’
- From Line 148 to 150, we have changed to ‘The silencing of CLA1 results in a bleached phenotype [35].’
- From Line 156 to 162, we have changed to ‘However, not all hirsutum varieties have an equal response to TRV-mediated silencing, which is an important consideration for future research on gene function [38]. Moreover, compared to G. hirsutum, the silencing of CLA1 is more prominent in G. arboretum and G. herbaceum, implying that the efficiency of TRV-mediated VIGS may be influenced by the ploidy level and appear to be higher in diploids.’
- From Line 165 to 171, we have changed to ‘Meanwhile, knocking down the expression of PSY in red leaf cotton resulted in green cotton leaves with no red color patches. These results suggested that PDS and PSY could be used as positive controls in the VIGS system. However, although the silencing of CLA1 and PDS can lead to visible photobleaching, it can also result in wilting and plant death. Therefore, it cannot be used for gene silencing throughout the entire plant growth period.’
- From Line 176 to 183, we have changed to ‘therefore, the PGF gene is a perfect marker for the entire cotton growth period. In a previous study, the proanthocyanidin (PA) metabolic-related genes ANS and anthocyanidin reductase (ANR) were also silenced via VIGS in barbadense. Transcripts of these two genes were dramatically reduced. The brownish phenotype of the cotton plants emerged from 8 to 10 days post-vaccination in the ANS-silenced lines, whereas no visible phenotype difference between the ANR-silenced lines and vector control plants was detected [37].’
- We have deleted Line 185 to 195.
- From Line 198 to 203, we have changed to ‘Abiotic stresses, such as drought, soil salinity and alkalinity, and extreme temperature stresses, can induce a series of physiological and biochemical reactions and inhibit normal growth and development of plants [41-43]. VIGS has been widely used to study gene function in cotton under abiotic stress (Table 1). The following sections list the application of VIGS to characterize abiotic-stress-related genes in cotton.’
- From Line 205 to 207, we have changed to ‘Drought is an important limiting factor in cotton production. Over half of the global cotton supply is grown in areas with severe water shortages. Cotton is recognized as drought-resistant.’
- From Line 243 to 244, we have changed to ‘due to its negative impact.’
- In Line 332, we have changed to ‘Verticillium is a serious disease caused by Verticillium dahliae’
- From Line 335 to 336, we have changed to ‘to improve the understanding of the molecular mechanism underlying cotton resistance to Verticillium.’
- In Line 339, we have changed to ‘’
- From Line 374 to 379, we have changed to ‘Whitefly has caused significant damage to global cotton production. Information on how cotton plants perceive and defend against the destructive pest is limited. RNA-seq was employed to compare two cotton cultivars exhibiting strong resistance and sensitivity to the whitefly. Analysis revealed that the transcriptional response of cotton to whitefly invasion involves transcription factors.’
- From Line 380 to 382, we have changed to ‘The virus-induced silencing of GhMPK3 in cotton led to the inhibition of the MPK-WRKY-jasmonic acid (JA) and ethylene (ET) pathways, and it increased susceptibility to the whitefly [64].’
- We have deleted Line 382 to 384.
- In Line 397, we have changed to ‘’
- We have deleted Line 401 to 403.
- From Line 405 to 412, we have changed to ‘Roots absorb water and nutrients supporting plants to respond to environmental stress and significantly affecting many important agronomic traits. However, cotton root structure is poorly understood. A GWAS of cotton root traits from 200 upland cotton accessions revealed two candidate genes, GhTRL1-A05 and GhPIN8-D04 [66]. The virus-induced silencing of these two genes in cotton resulted in a reduced root length and surface area.’ In addition, the abbreviation GWAS means genome-wide association study and has been listed in above Line 302.
- From Line 416 to 417, to avoid confusion, we have changed to ‘The virus-induced gene silencing of major DELLA genes particularly increased secondary cell wall (SCW).’
- From Line 421 to 422, to avoid confusion, we have deleted to ‘These findings provide a framework for GA to regulate the formation of SCW.’
- From Line 424 to 427, we have changed to ‘Knocking down the expression of GoSTR in J220 and Hai7124 through VIGS led to pubescent stems but no apparent leaf trichome changes.’
- From Line 429 to 430, we have changed to ‘Leaves are derived from the peripheral zone of SAM.’ In addition, SAM means shoot apical meristem and has been listed in above Line 414.
- From Line 443 to 446, we have changed to ‘GhLYI directly binds to the promoter of SENESCENCE-ASSOCIATED GENE 20 (SAG20) to activate its expression. The silencing of GhSAG20 via VIGS in cotton can delay leaf senescence [71].’
- We have deleted Line 448 to 452.
- From Line 453 to 455, we have changed to ‘The role of genes that regulate the development of anther and fertility are still yet to be determined.’
- From Line 455 to 459, we have changed to ‘The silencing of acyl-CoA N-acyltransferase (GhACNAT) through VIGS in cotton led to indehiscent anthers, reduced filaments and stamens, and plant sterility [72]. Further analysis revealed that GhACNAT played a critical role in controlling fertility by regulating lipid biosynthesis and the JA biogenesis pathways.’
- From Line 461 to 466, we have changed to ‘However, their exact role and potential regulatory mechanisms in cotton anther development are poorly understood. GhGPAT12/2 has been confirmed to regulate tapetum degradation, anther cuticle formation, and pollen wall development [73]. Further investigation showed that the expression of GhGPAT12/25 might be activated by GhMYB80s in the regulation of male fertility.’
- We have deleted Line 474 to 475.
- From Line 477 to 478, we have changed to ‘The process of cotton fiber development takes a long period of time, VIGS could also function in this process.’
- From Line 480 to 481, we have changed to ‘On the contrary, the suppression of WRINKLED1 expression led to.’
- From Line 484 to 488, we have changed to ‘A membrane lipid analysis was employed in wild-type cotton and fuzzless–lintless mutant fiber cells and ovules, and the content of phosphatidylinositol (PI) was found to be significantly higher in fiber cells.’
- From Line 495 to 497, we have changed to ‘The virus-induced gene silencing of these two genes could inhibit cotton fiber initiation, implying that they play crucial roles in fiber development in cotton. Phytohormones play vital roles in plant growth and development. However.’
- From Line 502 to 504, we have changed to ‘additionally, mature fiber cell wall thickness was apparently less than that of controls [79].’
- From Line 508 to 509, we have changed to ‘VIGS is one of the most promising and effective tools to investigate functional genes in cotton.’
- In Line 557, we have changed to ‘VIGS is a feasible method for quickly.’
- In Line 562, we have changed to ‘regulation of leaf development.’
- In Line 566, we have changed to ‘the green box indicates genes that respond to abiotic and biotic stresses.’
- From Line 567 to 569, we have changed to ‘Improvement in the inoculation method could expand the application of VIGS in cotton. A leaf injection is traditionally used. However.’
- In Line 574, we have changed to ‘Moreover, the efficient duration of virus-induced silencing and.’
- From Line 583 to 584, we have changed to ‘Therefore, it is a perfect indicator to trace the silencing efficiency through the entire growth period in cotton.’
- In Line 588 to 589, we have changed to ‘investigation of unknown gene function in cotton.’
- In Line 593, we have changed to ‘it has been proposed that.’
- From Line 595 to 597, to avoid confusion, we have deleted ‘At present, VIGS-based system has extended the gene toolbox beyond gene silencing, including virus-induced genome editing, overexpression, and host-induced gene silencing, which broaden the toolbox for non-model species [41].’
- In Line 598, we have changed to ‘These new findings.’
Reviewer 3 Report
Comments and Suggestions for Authors
A). Manuscript ID: Plants-2794620
B). Manuscript title: Applications of virus-induced gene silencing (VIGS) in cotton
C). Please revise the manuscript based on the following comments.
General Comments:
*Please follow the journal format, while revising the manuscript.
*Add scientific authority at the end of binomial names of all species, when they are mentioned for the first time in the manuscript.
*Include full forms of all the abbreviations/acronyms mentioned in the manuscript.
Specific Comments:
*Moderate to extensive English corrections required.
* Lines 12, 24to26, 172
*Line 175, 399, 401,525, and 531: Change 'leave' to 'leaf'
*Line 181: Change to 'soil salinity and alkalinity'
*Line 269: Change to 'effectively respond to'
*Line 353: Change to 'cotton led to'
*Line 396: Change to 'VIGS led into'
*Line 435: Change to 'dramatic reduction'
*Lines 445 and 446: Rewrite
*Line 448: Change to 'led to'
*Line 529: Change 'response' to 'respond'
*Line 557: Change to 'these new findings'
Comments on the Quality of English Language
*Extensive English corrections required throughout the manuscript.
*Please thoroughly proofread the manuscript, before resubmission.
Author Response
Thanks so much for the comments. We have revised the paper a lot.
Q1: Please follow the journal format, while revising the manuscript.
R1: We have revised the manuscript strictly followed with the journal format.
Q2: Add scientific authority at the end of binomial names of all species, when they are mentioned for the first time in the manuscript.
R2: We have revised the related part of the manuscript. From Line 82 to 85, we have changed to ‘such as Arabidopsis [10], tobacco (Nicotiana benthamiana) [11], wheat (Triticum aestivum) [12], alfalfa (Medicago truncatula) [13], tomato (Solanum lycopersicum) [14], and poplar (Populus euphratica) [15].’
Q3: Include full forms of all the abbreviations/acronyms mentioned in the manuscript.
R3: We have carefully checked the abbreviations/acronyms mentioned in the manuscript and revised through the whole review.
Q4: Moderate to extensive English corrections required.
R4: We have revised and improved English corrections a lot.
Q5: Lines 12, 24 to26, 172, Line 175, 399, 401,525, and 531: Change 'leave' to 'leaf'.
R5: We have revised the related part point by point through the whole manuscript.
Q6: Line 181: Change to 'soil salinity and alkalinity'.
R6: We have changed to ‘soil salinity and alkalinity’ in Line 198.
Q7: Line 269: Change to 'effectively respond to'
R7: We have changed to ‘effectively respond to’ in Line 291.
Q8: Line 353: Change to 'cotton led to'
R8: We have changed to ‘cotton led to’ in Line 380.
Q9: Line 396: Change to 'VIGS led into'
R9: We have changed to ‘VIGS led into’ in Line 425.
Q10: Line 435: Change to 'dramatic reduction'.
R10: We have changed to ‘dramatic reduction’ in Line 467.
Q11: Lines 445 and 446: Rewrite.
R11: We have rewrite from Line 477 to 478 ‘The process of cotton fiber development takes a long period of time, VIGS could also function in this process.’
Q12: Line 448: Change to ' led to '
R12: We have changed to ‘led to’ in Line 481.
Q13: Line 529: Change 'response' to 'respond'
R13: We have changed to ‘respond’ in Line 566.
Q14: Line 557: Change to 'these new findings'
R14: We have changed to ‘these new findings’ in Line 598.
Round 2
Reviewer 3 Report
Comments and Suggestions for Authors
Thank you, for revising the manuscript based on the suggestions.